# A Model for Accurate Determination of Environmental Parameters in Indoor Zoological and Botanical Gardens Supporting Efficient Species Management

León Latif Corral-Pesquera [1], Jonathan García-Manchón [1] and Pablo Morón-Elorza [2,3,*]

1    Rain Forest S.L., Biodomo—Parque de las Ciencias de Granada, Av de las Ciencias s/n, 18006 Granada, Spain
2    Department of Pharmacology and Toxicology, Faculty of Veterinary Medicine, Complutense University of Madrid, Av. Puerta de Hierro s/n, 28040 Madrid, Spain
3    Fundación Oceanogràfic de la Comunitat Valenciana, C/Eduardo Primo Yúfera (Científic) 1B, 46013 Valencia, Spain
*    Correspondence: p-moron@hotmail.com

**Abstract:** The detailed evaluation of environmental parameters can be a great tool for the optimal selection and location of vegetable species, not only in vegetable production facilities and greenhouses but also in zoological and botanical gardens, which frequently maintain delicate and exotic plant species with strict environmental requirements in immersive exhibits where conditions can vary remarkably. This study, developed at an indoor zoological garden (Biodomo—Parque de las Ciencias de Granada, Spain), evaluates a sampling protocol for the determination of seven environmental parameters: daily light integral (DLI) was determined at nine different locations of the facility using a portable Light Quantum SQ-500 sensor; air temperature, atmospheric pressure, and air relative humidity were measured using a fixed ATMOS14 sensor; and soil temperature, soil water content, and soil conductivity were determined using a fixed TEROS12 sensor. Values recorded for DLI showed statistically significant variations across the nine different sampling locations, as well as between the different months in all sampling spots. Significant variations were also detected across the 12 months of study for the rest of environmental parameters evaluated, and correlations were found between the studied parameters, with the correlation between soil and air temperature the strongest ($r_s = 0.758$) and soil temperature significantly superior to air temperature. The methodology described in this study can be easily reproduced in similar indoor zoological and botanical facilities, increasing the knowledge of the environmental conditions, and allowing corrections that could improve species selection, location, and management.

**Keywords:** daily light integral; conductivity; relative humidity; PAR light; air temperature; soil water content; ATMOS14; TEROS12

## 1. Introduction

The curiosity and desire to build artificial spaces that recreate nature within a safe environment for its contemplation and enjoyment has existed in our civilization for thousands of years. There are records of zoological and botanical gardens as far back as ancient Egypt [1]. Since the onset of long-distance travel and trade, humans have been fascinated by plant and animal species from other parts of the globe, so different from those present in their traditional environment. This has been accompanied by the desire to bring those foreign species back home to enjoy their natural properties, colors, and exotic shapes [2]. Transporting species from tropical areas to regions with colder climates indirectly demands ornamental greenhouses to allow their survival and development [3].

When focusing on vegetable species, the maintenance in indoor facilities such as botanical gardens has evolved enormously, from the first greenhouses that consisted of a simple protective cover from inclement weather to the sophisticated climate control systems

that currently exist, where practically all parameters can be controlled including lighting supplementation [4,5]. Plants within the zoological facilities play a fundamental role, since they act as vital support for the animals, enriching the space where they live in various ways and making it easier for these animals to enjoy a life that is as natural as possible across the 24 h period. Furthermore, they create an environment with natural vegetation specific to a certain region, and therefore help to "transport" the visitors to another habitat and provide a feeling of being immersed in a tropical destination [6].

Studies evaluating the effect of light on plant growth and development focus on photosynthetically active radiation (PAR), which corresponds to the spectral range of radiation comprised between 400 and 700 nanometers wavelength, and is the radiation used by photosynthetic organisms during photosynthesis [7]. The number of photosynthetically active particles of light (photons) of PAR, delivered to a specific area over a 24-h period, is studied as daily light integral (DLI). The variable DLI is frequently used to describe plant light requirements, and there are multiple studies determining DLI in different types of horticultural and ornamental crops [8–11]. However, there are currently no published studies determining DLI values in interior zoological facilities. Furthermore, the development of DLI determination studies has also proven of great use, not only in terrestrial species but also in aquatic animal and plant species [12]. The design of a habitat where plant and animal species coexist entails that a balance is reached. These habitats, in which animals successfully develop natural behaviors while plants thrive and resist animal erosion, require detailed planning and consideration of environmental parameters [13]. In interior facilities, an environmental parameters design is essential for adequate species choice and distribution; if optimal lighting, temperature, humidity, etc., are not provided, plants will not develop and grow successfully, and the desired aesthetic potential will not be achieved [14]. Considering this, the quantification of light input and environmental parameters such as relative humidity, air, and soil temperature determination acquire a great importance, not only in vegetable production units but also in zoological facilities, in order to evaluate which lighting and environmental installation will be necessary, as well as the effective selection of species based on their environmental requirements [12,14].

The use of environmental sensors for the measurement of environmental parameters is already carried out in indoor horticultural and livestock farms; this has demonstrated the importance of environmental parameter (PAR light, temperature, air relative humidity, etc.) determination for the accurate management of these facilities [15,16]. On the other hand, studies determining environmental parameters in zoological and botanical facilities are still very limited, with little attention paid to the selection, use and management of the vegetable species, and with few studies reporting the importance of certain environmental parameters for the optimal health and development of animals [6,17–19]. In livestock and horticultural production facilities, environmental parameter control studies are mainly focused on increasing production and nutritional quality, while in zoological and botanical gardens the main objective of environmental studies should be to improve animal and plant welfare, as well as to generate better criteria for species selection based on these parameters [15,16,20].

Because of this, the objective of our study was to develop and describe a method for environmental parameters measurement for indoor zoological gardens, combining different sensors of scientific quality used in indoor horticultural and livestock farms as well as research centers, which could be used easily and safely in mixed indoor installations in which plants and animals coexist. Our study aims to evaluate the potential use of these sensors and the sampling model developed, to obtain baseline data on the main environmental parameters that could provide information to improve and help adjust the climate and lighting control systems of this type of facility, as well as the selection, location and management of both plant and animal species within it.

## 2. Materials and Methods

### 2.1. Environmental Conditions

This study was developed at Biodomo—Parque de las Ciencias de Granada (Av. de las Ciencias. s/n. Granada, Spain; https://www.parqueciencias.com/biodomo (accessed on 10 September 2022)). Biodomo is a zoological and botanical facility inaugurated in 2016, which recreates terrestrial and aquatic habitats of the equatorial and tropical regions of the globe (Figure 1). Terrestrial ecosystems correspond to the Amazon rainforest, Southeast Asia and Madagascar; aquatic ecosystems represent fluvial, mangrove, coastal, and marine ecosystems. The present study focuses on the terrestrial multispecies enclosures of Biodomo, which house different animal and plant species living together, creating an ecosystem where they coexist in balance. With a total volume of 18,000 m$^3$, a total surface of 4252 m$^3$ and total exhibit surface of 2700 m$^2$, Biodomo is an interior zoological and botanical facility where the contribution of natural light through the roof is through polycarbonates, which constitute 30% of the roof surface. The rest of the roof is constituted by metal and thermal isolation where light does not penetrate.

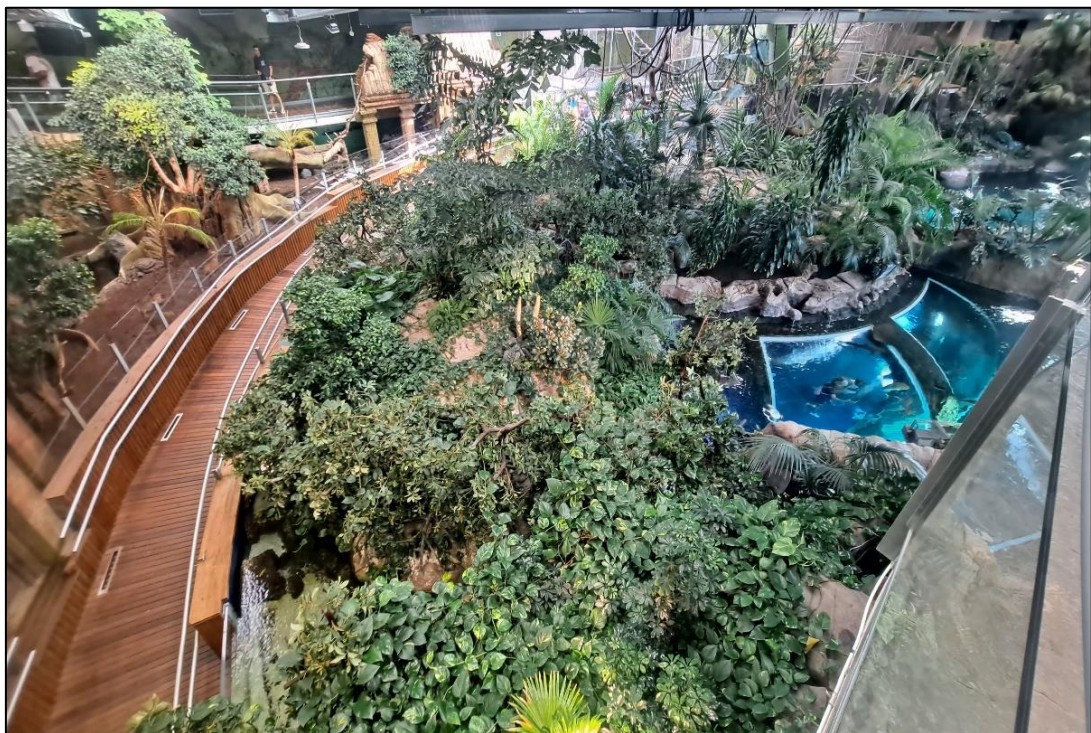

**Figure 1.** Biodomo—Parque de las Ciencias de Granada. Indoor facilities.

To enrich the natural light input, lighting is supplemented by light-emitting diode (LED) screens (Sequoia Cultiva Wall 50–60 Hz 224 W IGNIA GREEN led screens), which emit photosynthetic active radiation (PAR) light spectrum as well as white light LED screens (Konak 200 W 5000 K 150° IP65 SECOM) with 12 h light: 12 darkness programmed periods The location of the PAR and white light LED screens throughout Biodomo has been optimized during the last four years and is represented in Figure 2. Environmental parameters at Biodomo are regulated by an air conditioning system made up of two water-water chillers and two air treatment units. Air treatment units purify the air using fans and filters (consisting of G4 Class Air 10 micron filter, carbon filters, and F7 EU7 Class Air filter which are renewed periodically), as well as climatize by heating, or refrigerating the air to achieve the desired temperature depending on the season. Air humidity is maintained using a fogging system controlled by a humidistat (SMD4500 controller; ELLIWELL Ibérica; 46980 Valencia, Spain) Furthermore, the facility also has several additional auxiliary systems for climate control. For instance, some areas of Biodomo are equipped with fabric awnings

on the ceiling below the roof that can be deployed to create shadows in the summer months when solar radiation is very strong. The roof of the Biodomo also has two movable skylights, which can be opened to increase sunlight input and air exchange when exterior conditions are optimal for animals and plants.

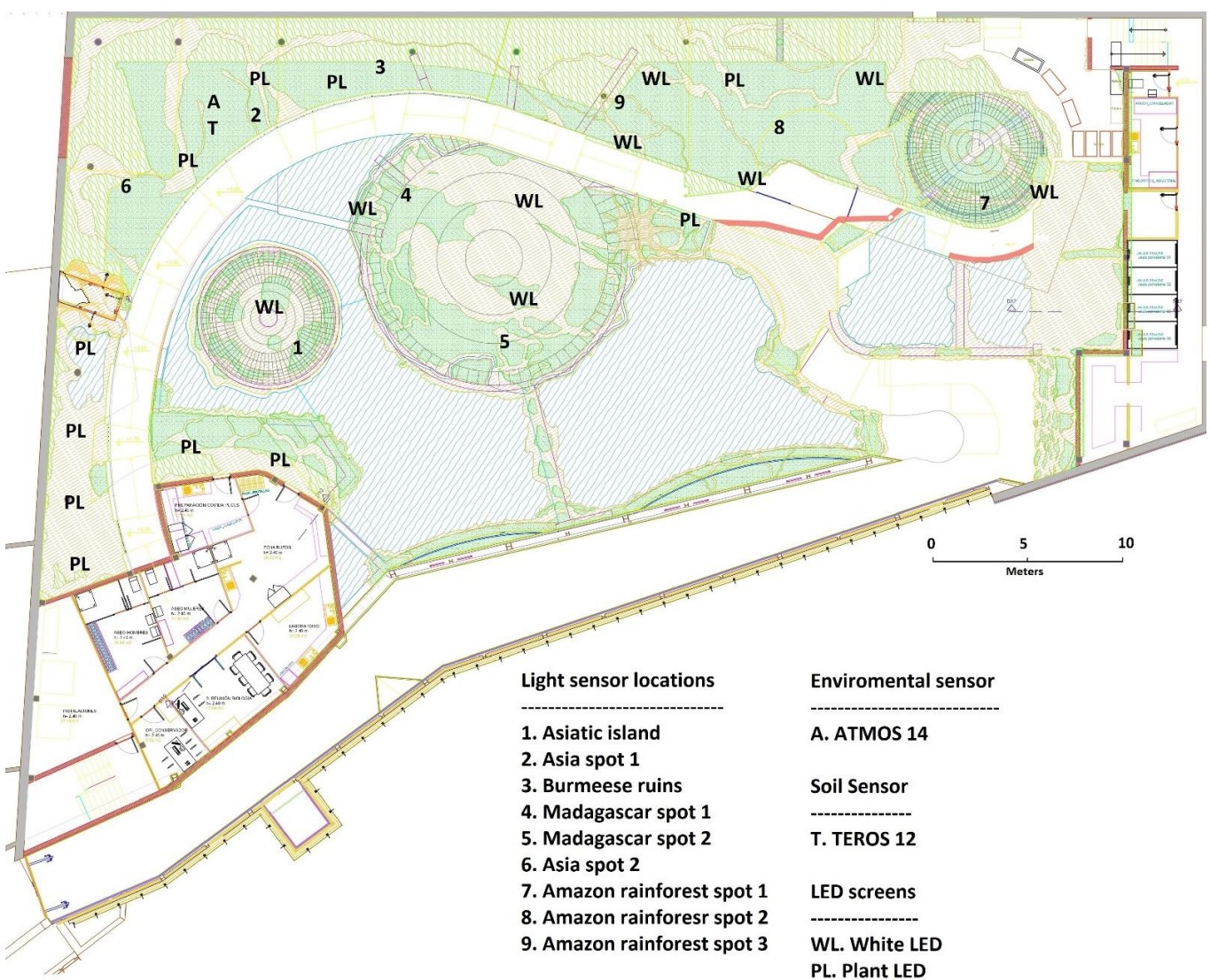

**Light sensor locations**
-------------------------------
1. Asiatic island
2. Asia spot 1
3. Burmeese ruins
4. Madagascar spot 1
5. Madagascar spot 2
6. Asia spot 2
7. Amazon rainforest spot 1
8. Amazon rainforesr spot 2
9. Amazon rainforest spot 3

**Enviromental sensor**
-------------------------------
A. ATMOS 14

**Soil Sensor**
---------------
T. TEROS 12

**LED screens**
---------------
WL. White LED
PL. Plant LED

**Figure 2.** Representative map of Biodomo and sensor location. Please note that the light sensor was rotated across nine different positions within Biodomo, while environmental sensors (ATMOS 14 and TEROS 12 sensors) were located in a fixed sensor placement.

### 2.2. Sensors

### 2.2.1. Light Sensor

For this study, one Light Quantum Sensor Model SQ-500 (Apogee Instruments. Inc., Logan, UT 84321, USA) was used. Sensor sensitivity was 0.01 mV/µmol·m$^{-2}$s$^{-1}$, measurement range was 0 to 4000 µmol m$^{-2}$s$^{-1}$, long-term drift (non-stability) was under 2% per year, non-linearity less than 1%, and response time less than 1 ms, a field of view 180°, spectral range 389 to 692 nm $\pm$ 5 nm, spectral selectivity less than 10% from 412 to 682 $\pm$ 5 nm, azimuth error less than 0.5%, tilt error less than 0.5%, and temperature response $-0.11 \pm 0.04$% per °C. This sensor registered one DLI reading every 3 min, and the daily values were computed as the total DLI per squared meter per day.

To increase the reliability of the data obtained and determine the DLI that the plants within the facility were being exposed to, the light sensor was located in the enclosure at

the vegetation level. As the different animal species maintained at the enclosure could interact with the sensor throughout the day, and in order to protect it, a protective cage was designed using wood on the sides and base, and a 150 mm methacrylate lid was fixed to the cage using a nylon flange (Figure 3). While most avian species seemed not affected by the introduction of the sensor in their enclosures, some mammal species showed an initial interest in the sensor protective cage, such as ring-tailed lemurs (*Lemur catta*), the white-faced saki (*Pithecia pithecia*), and the white-lipped tamarin (*Saguinus labiatus*). These animals are provided weekly with different enrichments and initially inspected the cage in detail for food, though animals did not show further interest after the first week and during further data collections throughout the 12-month study.

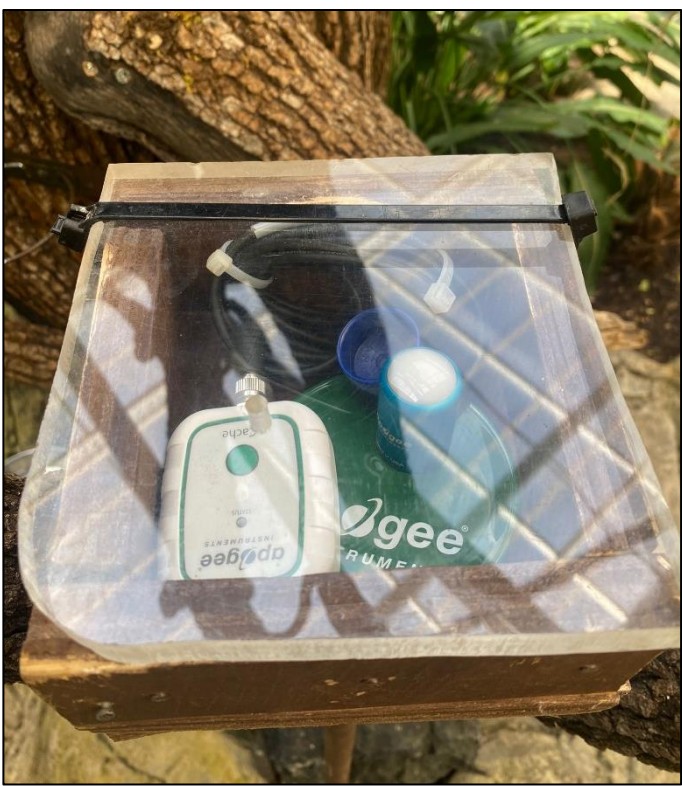

**Figure 3.** Detail of the light sensor placed inside the wooden tailor-made protective cage and the methacrylate protective cover. Biodomo—Parque de las Ciencias de Granada.

The photosynthetic active radiation (PAR) light (measured in photosynthetic photon flux density (PPFD)) loss percentage associated with the protective cage was calculated by taking ten repetitive readings, with and without the cage, to determine the light loss percentage caused by the protective cage, which resulted in a mean loss of 8.18% $\pm$ 0.53 SD (min 7.42%, max 9.28%). The PPFD results provided in this study have been expressed as corrected data.

### 2.2.2. Air and Soil Sensors

For temperature, ambient humidity, and atmospheric pressure measurement, this study used one ATMOS14 sensor (Meteor group, Inc., 2365 NE Hopkins Court, Pullman, WA 99163, USA). Temperature measurement range was $-40$ to $80\ ^\circ$C, resolution of $\pm0.1\ ^\circ$C, accuracy $\pm0.5\ ^\circ$C, equilibration time <400 s, long-term drift <0.004 $^\circ$C/year. Relative Humidity (RH) measurement range was 0 to 100% RH (0.00–1.00), resolution was 0.1% RH, equilibration time <40 s, hysteresis <1% RH, long-term drift 0.5% RH/year. Barometric pressure measurement range was 50 to 110 kPa, resolution of 0.01 kPa, accuracy of $\pm0.4$ kPa.

For soil temperature, water content (Volumetric Water Content (VWC), and salt concentration (Bulk Electrical Conductivity (EC)) measurement, we have used one TEROS

12 sensor (Meteor group, Inc., 2365 NE Hopkins Court, Pullman, WA 99163, USA). This sensor had a temperature measurement range of −40 to +60 °C, resolution of ±0.1 °C, accuracy of ±0.5 °C from −40 to 0 °C, and ±0.3 °C from 0 to +60 °C. The VWC sensor had a mineral soil calibration of 0.00–0.70 m$^3$/m$^3$, soilless media calibration of 0.0–1.0 m$^3$/m$^3$, apparent dielectric permittivity (εa) 1 (air) to 80 (water), resolution of 0.001 m$^3$/m$^3$, apparent dielectric permittivity (εa) of 1–40 (soil range), ±1 εa (unitless) 40–80, 15% of measurement, bulk EC measurement range of 0 to 20 dS/m (bulk), resolution of 0.001 dS/m, and accuracy of ±(5% + 0.01 dS/m) (Figure 4).

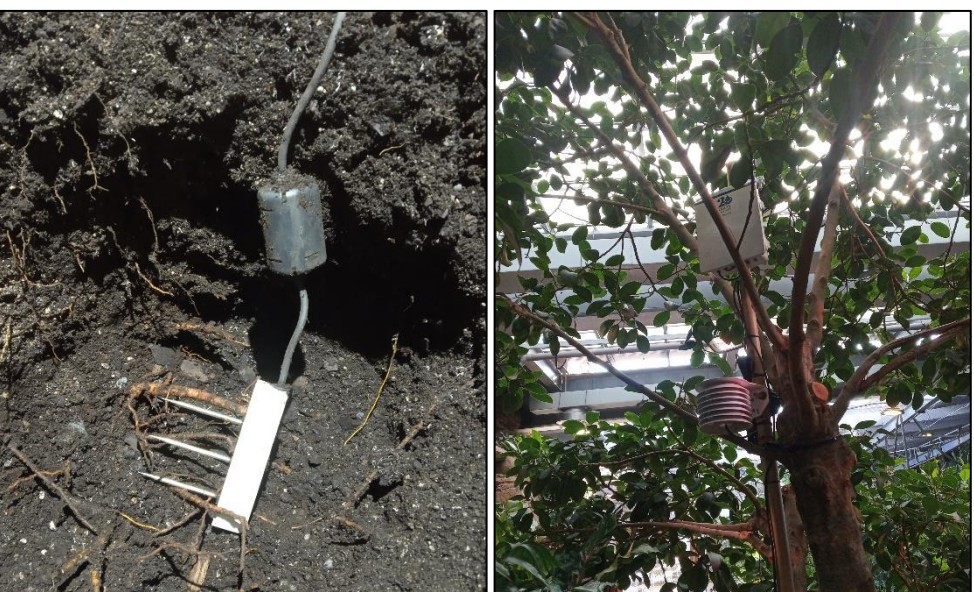

**Figure 4.** Detail of the ATMOS14 (**left**) and TEROS12 (**right**) probes. Biodomo—Parque de las Ciencias de Granada.

### 2.3. Data Collection

2.3.1. Light (DLI) Sampling Protocol

To carry out this study, nine sampling points were selected for DLI determination. These sampling points are represented in Figure 2 and were designated because they are representative of the different enclosures and vegetation environments of Biodomo. The sensor, which registered one reading every 3 min, determined a total DLI value per square meter per day; it was moved daily across the different sampling locations from point number 1 to 9, thus rotating through all the points from 1 January 2021 to 31 December 2021 with a 9-day cycle and for 365 days. This portable data logger and the designed rotating system allowed us to obtain three to four monthly DLI measurements at each sampling point and then averaged these measurements to obtain the monthly DLI.

2.3.2. Sampling Protocol for the Rest of the Environmental Parameters

As represented in Figure 2, ATMOS14 and TEROS12 probes were located in a fixed position at the Asian installation at Biodomo and were not rotated throughout the year like the light sensor. The ATMOS14 sensor was located hidden between the foliage of a Port Jackson fig (*Ficus rubiginosa*) at 3 m above the ground, and the TEROS12 sensor was buried at a depth of 50 cm, under the same tree and between the roots. While the light sensor was portable, the data logger shared between ATMOS14 and TEROS12 was non-portable, and the strategic point was determined for the placement of both sensors. While light depends on the structure of the building and varies across the different areas and enclosures, the air environmental parameters collected by the ATMOS14 sensor are representative of the entire indoor facility. The soil probe TEROS12 was not translocated either, and its fixed location was maintained throughout the study as it was buried in the ground and lacked a portable datalogger. Both sensors were configured to register measures every 10 min. Data

recording started on 27 May 2021 and ended on 25 May 2022; these dates correspond to the availability of the sensor for sampling at Biodomo.

### 2.4. Data and Statistical Analysis

The distribution of the DLI data was evaluated using a Kolmogorov–Smirnov normality test with Lilliefors correction, which showed that the data followed a non-Gaussian distribution [21]. To evaluate the existence of significant differences between the different sampling locations and between the different months at each sampling point, a Kruskal–Wallis test and Dunn's post hoc tests with Bonnferroni corrections were used, and significance was set at a *p*-value of 0.05. Statistical comparison between air and soil temperature was performed using a Mann–Whitney U test [22]. Correlation between the different environmental parameters was evaluated by calculating the Spearman's rank correlation coefficient ($r_s$) with a confidence interval of 0.95. A principal component analysis (PCA) was performed for the six variables measured using the ATMOS14 and TEROS12 sensors; daily light integral analysis has not been included in the PCA model due to a different sampling methodology. All statistical analyses were performed using the statistical software package RStudio R (Version 1.2.504; RStudio Team. 2020 Boston, MA 02210, USA; www.rstudio.com (accessed on 20 September 2022)).

## 3. Results

### 3.1. Variations in DLI

Monthly DLI values and descriptive statistics for the different sampling locations are reported in Table 1. Values recorded for DLI showed statistically significant variations across the different sampling locations, as well as between the different months. Statistically significant differences ($p < 0.05$; Dunn's post hoc with Bonnferroni corrections) were detected between the DLI values recorded in the months of June vs. December, June vs. January, June vs. November, July vs. November, May vs. November, May vs. December, and May vs. January. Furthermore, as it can be appreciated in Figure 5, important differences were detected for DLI annual measurements between the locations (Figure 5). These differences were statistically significant (Dunn's post hoc with Bonnferroni corrections) between: Amazon rainforest spot 1 vs. Amazon rainforest spot 2 ($p < 0.05$), Amazon rainforest spot 1 vs. Amazon rainforest spot 2 ($p < 0.05$), Amazon rainforest spot 3 vs. Asiatic island ($p < 0.01$), Amazon rainforest spot 1 vs. Asia spot 1 ($p < 0.05$), Amazon rainforest spot 2 vs. Asiatic Island ($p < 0.01$), Asia spot 1 vs. Asiatic island ($p < 0.01$), Asia enclosure spot 2 vs. Asiatic island ($p < 0.05$), Madagascar enclosure spot 1 vs. Asiatic island ($p < 0.01$), Burmese ruins vs. Asiatic island ($p < 0.01$), Amazon rainforest spot 3 vs. Madagascar enclosure spot 2 ($p < 0.01$), Amazon rainforest spot 2 vs. Madagascar enclosure spot 2 ($p < 0.01$), Asia spot 1 vs. Madagascar enclosure spot 2 ($p < 0.01$), Amazon rainforest spot 1 vs. Burmeese ruins ($p < 0.05$), Madagascar enclosure spot 2 vs. Burmeese ruins ($p < 0.05$), and Madagascar enclosure spot 1 vs. Madagascar enclosure spot 2 ($p < 0.05$). Statistical comparison between the different months across the sampling spots was not possible due to a small number of samples per month and spot (3–4 samples) associated with the rotating system and limited by the availability of only one DLI sensor.

**Table 1.** Descriptive statistics for the DLI measurements (mol/m$^2$day) in the nine different sampling locations across the different months.

| Sampling Spot | Statistics | January | February | March | April | May | June | July | August | September | October | November | December |
|---|---|---|---|---|---|---|---|---|---|---|---|---|---|
| Asiatic Island | MEAN | 0.3 | 0.6 | 0.8 | 0.6 | 0.4 | 0.6 | 0.7 | 0.4 | 0.4 | 0.7 | 1.4 | 0.6 |
| | MEDIAN | 0.3 | 0.7 | 0.6 | 0.6 | 0.4 | 0.6 | 0.6 | 0.4 | 0.4 | 0.5 | 1.7 | 0.7 |
| | SD | 0.1 | 0.3 | 0.2 | 0.3 | 0.2 | 0.2 | 0.0 | 0.1 | 0.1 | 0.5 | 0.6 | 0.3 |
| | N | 4.0 | 3.0 | 3.0 | 4.0 | 3.0 | 3.0 | 3.0 | 3.0 | 3.0 | 3.0 | 3.0 | 3.0 |
| | MIN | 0.2 | 0.3 | 0.6 | 0.3 | 0.3 | 0.4 | 0.6 | 0.4 | 0.4 | 0.3 | 0.8 | 0.3 |
| | MAX | 0.5 | 0.8 | 1.0 | 0.9 | 0.6 | 0.8 | 0.7 | 0.5 | 0.5 | 1.3 | 1.8 | 0.9 |
| Asia 1 | MEAN | 1.4 | 2.9 | 7.6 | 4.5 | 9.8 | 14.6 | 15.6 | 9.8 | 5.6 | 3.6 | 1.8 | 1.9 |
| | MEDIAN | 1.4 | 3.2 | 8.0 | 4.5 | 9.7 | 14.5 | 15.3 | 10.2 | 5.8 | 3.4 | 1.7 | 1.9 |
| | SD | 0.4 | 0.5 | 1.9 | 2.1 | 2.0 | 2.5 | 2.0 | 3.0 | 0.9 | 0.6 | 0.3 | 0.1 |
| | N | 4.0 | 3.0 | 4.0 | 2.0 | 3.0 | 3.0 | 4.0 | 3.0 | 4.0 | 3.0 | 3.0 | 3.0 |
| | MIN | 1.1 | 2.4 | 5.0 | 3.0 | 7.9 | 12.1 | 13.4 | 6.6 | 6.6 | 3.2 | 1.4 | 1.8 |
| | MAX | 2.0 | 3.4 | 9.3 | 5.9 | 11.9 | 17.1 | 18.2 | 12.5 | 12.5 | 4.3 | 2.1 | 2.0 |
| Burmeese ruins | MEAN | 2.3 | 4.3 | 8.3 | 4.3 | 13.7 | 13.9 | 14.1 | 11.0 | 5.5 | 2.3 | 1.2 | 1.7 |
| | MEDIAN | 2.4 | 4.8 | 8.3 | 3.6 | 10.8 | 14.1 | 14 | 11.7 | 6.8 | 2.3 | 1.3 | 1.8 |
| | SD | 0.5 | 2.3 | 1.1 | 2.0 | 5.7 | 0.5 | 2.7 | 2.9 | 2.6 | 0.3 | 0.6 | 0.1 |
| | N | 3.0 | 4.0 | 2.0 | 3.0 | 3.0 | 3.0 | 4.0 | 4.0 | 3.0 | 3.0 | 4.0 | 3.0 |
| | MIN | 1.7 | 1.1 | 7.6 | 2.8 | 10.1 | 13.4 | 11.6 | 7.3 | 7.3 | 1.9 | 0.4 | 1.6 |
| | MAX | 2.7 | 6.5 | 9.1 | 6.6 | 20.3 | 14.3 | 16.9 | 13.6 | 13.6 | 2.5 | 1.7 | 1.8 |
| Madagascar 1 | MEAN | 2.0 | 2.5 | 3.7 | 7.5 | 21.1 | 7.6 | 8.1 | 4.2 | 3.5 | 2.4 | 1.9 | 1.5 |
| | MEDIAN | 2.1 | 2.6 | 3.0 | 7.4 | 21.1 | 9.4 | 8.1 | 4.2 | 3.4 | 2.4 | 2.0 | 1.6 |
| | SD | 0.3 | 0.5 | 1.4 | 1.1 | 0.8 | 4.7 | 1.8 | 0.0 | 0.5 | 0.5 | 0.3 | 0.2 |
| | N | 3.0 | 3.0 | 4.0 | 3.0 | 2.0 | 3.0 | 2.0 | 1.0 | 3.0 | 4.0 | 4.0 | 4.0 |
| | MIN | 1.7 | 1.9 | 2.9 | 6.4 | 20.6 | 2.3 | 6.8 | 4.2 | 4.2 | 1.9 | 1.6 | 1.4 |
| | MAX | 2.2 | 2.9 | 5.7 | 8.7 | 21.6 | 11.3 | 9.3 | 4.2 | 4.2 | 3.1 | 2.2 | 1.8 |
| Madagascar 2 | MEAN | 0.3 | 0.5 | 0.8 | 1.2 | 1.8 | 1.7 | 1.7 | 1.1 | 1.1 | 0.9 | 0.6 | 0.8 |
| | MEDIAN | 0.3 | 0.5 | 0.8 | 1.3 | 1.7 | 1.7 | 1.7 | 1.1 | 1.1 | 0.9 | 0.6 | 0.7 |
| | SD | 0.1 | 0.2 | 0.2 | 0.2 | 0.3 | 0.1 | 0.1 | 0.5 | 0.1 | 0.1 | 0.1 | 0.2 |
| | N | 3.0 | 3.0 | 4.0 | 3.0 | 3.0 | 3.0 | 3.0 | 2.0 | 4.0 | 4.0 | 4.0 | 3.0 |
| | MIN | 0.2 | 0.3 | 0.5 | 1.0 | 1.5 | 1.6 | 1.6 | 0.8 | 0.8 | 0.9 | 0.4 | 0.6 |
| | MAX | 0.4 | 0.6 | 1.0 | 1.4 | 2.2 | 1.8 | 1.7 | 1.5 | 1.5 | 1.0 | 0.7 | 1.0 |
| Asia 2 | MEAN | 1.0 | 1.7 | 2.3 | 2.4 | 3.4 | 2.8 | 2.9 | 2.7 | 1.8 | 1.3 | 0.9 | 0.9 |
| | MEDIAN | 1.0 | 1.7 | 2.3 | 2.4 | 3.5 | 3.0 | 2.7 | 2.9 | 1.8 | 1.2 | 0.9 | 1.0 |
| | SD | 0.2 | 0.1 | 0.4 | 0.4 | 2.8 | 0.6 | 0.5 | 0.5 | 0.2 | 0.2 | 0.0 | 0.2 |
| | N | 3.0 | 2.0 | 4.0 | 3.0 | 3.0 | 3.0 | 3.0 | 3.0 | 2.0 | 4.0 | 1.0 | 3.0 |
| | MIN | 0.8 | 1.6 | 1.7 | 1.9 | 3.2 | 2.2 | 2.6 | 2.2 | 2.2 | 1.2 | 0.9 | 0.7 |
| | MAX | 1.3 | 1.7 | 2.6 | 2.8 | 4.8 | 3.2 | 3.4 | 3.1 | 3.1 | 1.6 | 0.9 | 1.1 |
| Amazon 1 | MEAN | 0.7 | 0.9 | 1.2 | 2.1 | 2.3 | 3.0 | 2.3 | 1.4 | 1.0 | 1.0 | 0.7 | 0.7 |
| | MEDIAN | 0.7 | 0.9 | 1.1 | 2.1 | 2.5 | 3.0 | 2.4 | 1.5 | 1.2 | 1.0 | 0.7 | 0.7 |
| | SD | 0.2 | 0.1 | 0.3 | 0.4 | 0.7 | 1.0 | 0.7 | 0.3 | 0.4 | 0.0 | 0.0 | 0.1 |
| | N | 3.0 | 3.0 | 3.0 | 4.0 | 3.0 | 2.0 | 4.0 | 3.0 | 3.0 | 3.0 | 2.0 | 4.0 |
| | MIN | 0.6 | 0.8 | 1.0 | 1.7 | 1.6 | 2.3 | 1.3 | 1.1 | 1.1 | 1.0 | 0.6 | 0.6 |
| | MAX | 1.0 | 1.0 | 1.6 | 2.7 | 2.9 | 3.6 | 2.9 | 1.6 | 1.6 | 1.0 | 0.7 | 0.8 |
| Amazon 2 | MEAN | 1.7 | 3.9 | 10.9 | 10.6 | 10.6 | 11.1 | 10.7 | 7.6 | 4.7 | 2.2 | 1.6 | 1.0 |
| | MEDIAN | 1.5 | 4.5 | 14.1 | 10.5 | 10.6 | 11.1 | 11.1 | 7.5 | 4.8 | 2.2 | 1.6 | 1.1 |
| | SD | 0.6 | 1.5 | 7.2 | 5.5 | 1.1 | 0.4 | 1.3 | 0.2 | 0.2 | 2.0 | 0.3 | 0.1 |
| | N | 3.0 | 3.0 | 3.0 | 4.0 | 2.0 | 2.0 | 4.0 | 3.0 | 2.0 | 2.0 | 2.0 | 4.0 |
| | MIN | 1.2 | 2.2 | 2.6 | 4.9 | 9.8 | 10.8 | 8.9 | 7.4 | 7.4 | 0.8 | 1.4 | 0.9 |
| | MAX | 2.3 | 5.0 | 16.0 | 16.2 | 11.4 | 11.4 | 11.7 | 7.8 | 7.8 | 3.6 | 1.9 | 1.2 |
| Amazon 3 | MEAN | 1.6 | 2.2 | 6.5 | 7.0 | 14.2 | 11.6 | 7.5 | 6.7 | 5.6 | 4.0 | 1.6 | 1.1 |
| | MEDIAN | 1.6 | 2.1 | 5.1 | 7.1 | 15.1 | 12.4 | 7.5 | 7.5 | 6.4 | 4.7 | 1.6 | 1.1 |
| | SD | 0.1 | 0.7 | 4.8 | 3.8 | 3.7 | 2.6 | 0.4 | 2.3 | 2.3 | 1.5 | 0.4 | 0.0 |
| | N | 3.0 | 3.0 | 3.0 | 4.0 | 3.0 | 3.0 | 2.0 | 3.0 | 3.0 | 3.0 | 2.0 | 3.0 |
| | MIN | 1.5 | 1.6 | 2.6 | 3.1 | 10.2 | 8.7 | 7.2 | 4.1 | 4.1 | 2.2 | 1.3 | 1.1 |
| | MAX | 1.7 | 2.9 | 11.9 | 10.8 | 17.3 | 13.6 | 7.8 | 8.5 | 8.5 | 5.0 | 1.9 | 1.1 |

N = number of days sampled per month.

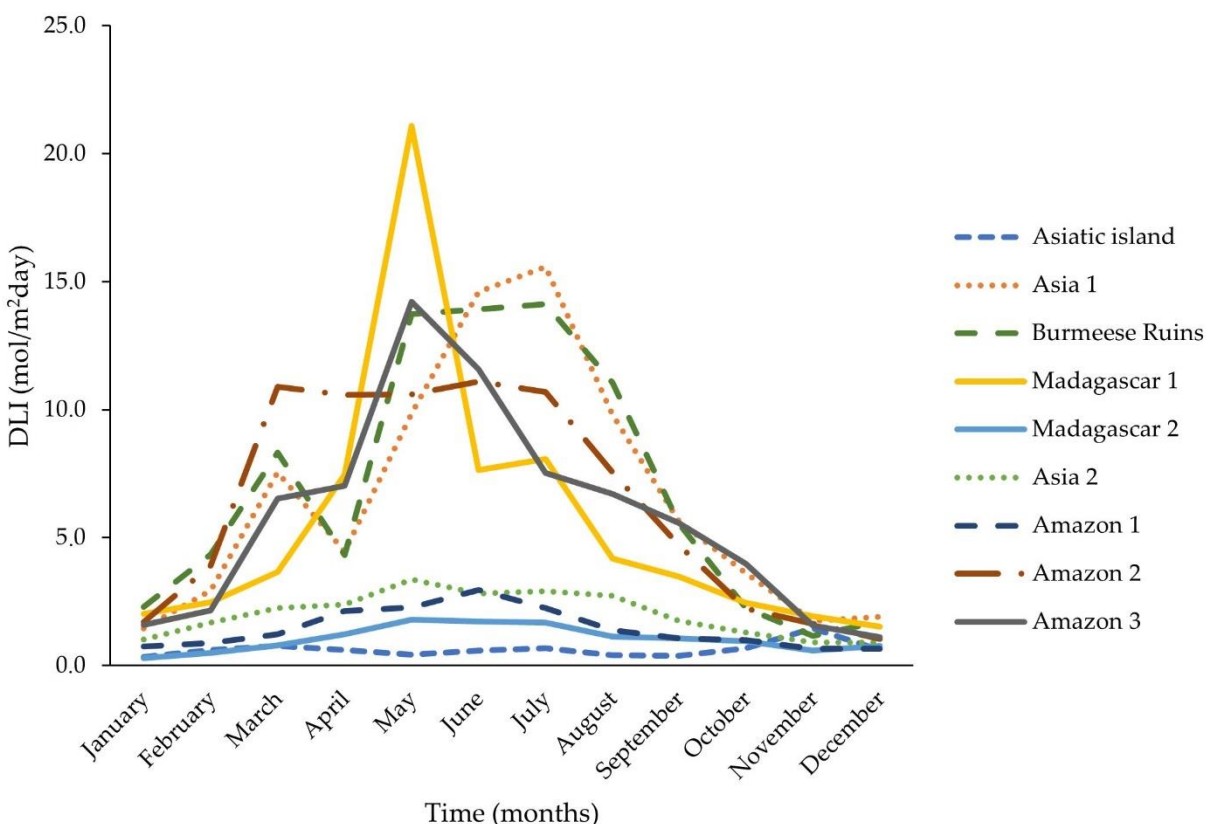

**Figure 5.** Daily light integral (DLI) mean values for the different months across sampling spots at Biodomo—Parque de las Ciencias de Granada. A Light Quantum Sensor Model SQ-500 (Apogee Instruments. Inc., Logan, UT 84321, USA) was used, which determined the total daily DLI, based on individual DLI values registered every 3 min. The DLI sensor was moved daily across the nine different sampling locations, with a nine-day cycle and for 365 days (1 January 2021 to 31 December 2021).

*3.2. Variations in Air Temperature, Air Relative Humidity, Ambient Pressure, Soil Water Content, Soil Temperature and Conductivity*

Descriptive statistics for air temperature (°C), relative humidity (RH), ambient pressure (kPa), soil water content ($m^3/m^3$), soil temperature (°C), and conductivity (mS/cm) at the fixed sampling spot for the different months are reported in Table 2. When focusing on air temperature, statistically significant differences ($p < 0.01$; Dunn's post hoc with Bonnferroni corrections) were detected between all months with the exception of December vs. November ($p = 0.095$) and April vs. February ($p = 0.205$); no differences were detected between August and July ($p = 1.000$). For relative humidity, statistically significant differences ($p < 0.01$; Dunn's post hoc with Bonnferroni corrections) were detected between all sampled months except for: November vs. September ($p = 0.051$), January vs. September ($p = 0.016$), November vs. October (0.019), May vs. October ($p = 0.515$), and April vs. October ($p = 0.166$). No differences were detected between May and November, February vs. November, February vs. March, August vs. March, and August vs. February ($p = 1.000$). Something similar happened with atmospheric pressure, and statistically significant differences ($p < 0.01$; Dunn's post hoc with Bonnferroni corrections) were detected between all months except for July vs. November ($p = 0.193$). Soil water content also showed statistically significant differences ($p < 0.01$; Dunn's post hoc with Bonnferroni corrections) between all studied months except for August vs. September ($p = 0.382$). The same happened with soil temperature, and all months showed statistically significant differences ($p < 0.01$; Dunn's post hoc with Bonnferroni corrections) except for August vs. July ($p = 0.055$) and December vs. February ($p = 0.087$). Conductivity also varied significantly ($p < 0.01$; Dunn's post hoc with Bonnferroni corrections) across the different months, except for July vs. November ($p = 0.158$) and June vs. November ($p = 0.035$). Values for the main

environmental parameters recorded every 10 min by ATMOS14 and TEROS12 probes are represented in Figure 6.

**Table 2.** Descriptive statistics for the main environmental parameters measured using a fixed ATMOS and TEROS sample probe.

| Parameter (Unit) | Statistic | January | February | March | April | May | June | July | August | September | October | November | December |
|---|---|---|---|---|---|---|---|---|---|---|---|---|---|
| Air temperatura (°C) | MEAN | 21.91 | 22.55 | 22.72 | 22.63 | 23.74 | 24.46 | 25.47 | 25.47 | 24.35 | 23.25 | 22.28 | 22.17 |
| | MEDIAN | 22.00 | 22.30 | 22.40 | 22.37 | 23.65 | 24.52 | 25.50 | 25.70 | 24.50 | 23.20 | 22.20 | 22.20 |
| | SD | 0.44 | 0.90 | 0.72 | 0.82 | 1.20 | 1.23 | 0.96 | 1.08 | 0.91 | 0.88 | 0.64 | 0.43 |
| | MAX | 23.60 | 25.10 | 24.90 | 25.16 | 27.30 | 28.08 | 28.10 | 29.10 | 26.90 | 25.70 | 24.91 | 24.00 |
| | MIN | 19.20 | 17.15 | 20.31 | 17.60 | 20.64 | 19.80 | 21.70 | 19.98 | 20.40 | 19.82 | 19.81 | 20.20 |
| | N | 2035 | 4025 | 4458 | 4315 | 3533 | 4317 | 4462 | 4463 | 4320 | 4464 | 4320 | 4464 |
| Relative Humidity (%) | MEAN | 0.64 | 0.63 | 0.64 | 0.65 | 0.65 | 0.62 | 0.61 | 0.63 | 0.65 | 0.66 | 0.66 | 0.67 |
| | MEDIAN | 0.66 | 0.64 | 0.64 | 0.67 | 0.67 | 0.63 | 0.62 | 0.65 | 0.66 | 0.67 | 0.67 | 0.68 |
| | SD | 0.06 | 0.07 | 0.05 | 0.06 | 0.07 | 0.06 | 0.06 | 0.07 | 0.05 | 0.05 | 0.07 | 0.04 |
| | MAX | 0.94 | 0.91 | 0.86 | 0.88 | 0.90 | 0.89 | 0.88 | 0.87 | 0.85 | 0.94 | 0.82 | 0.84 |
| | MIN | 0.50 | 0.32 | 0.44 | 0.32 | 0.41 | 0.31 | 0.35 | 0.30 | 0.43 | 0.38 | 0.41 | 0.55 |
| | N | 4454 | 4025 | 4458 | 4315 | 3533 | 4317 | 4462 | 4463 | 4320 | 4464 | 4320 | 4464 |
| Atmospheric pressure (kPa) | MEAN | 94.88 | 94.72 | 93.92 | 93.83 | 94.19 | 94.13 | 94.09 | 94.08 | 94.23 | 94.36 | 94.08 | 94.58 |
| | MEDIAN | 94.92 | 94.76 | 93.97 | 93.76 | 94.26 | 94.12 | 94.09 | 94.07 | 94.19 | 94.35 | 94.14 | 94.68 |
| | SD | 0.31 | 0.32 | 0.49 | 0.48 | 0.32 | 0.25 | 0.19 | 0.22 | 0.24 | 0.24 | 0.37 | 0.35 |
| | MAX | 95.56 | 95.31 | 95.08 | 94.76 | 94.73 | 94.89 | 94.61 | 94.65 | 94.91 | 94.96 | 95.03 | 95.20 |
| | MIN | 93.83 | 93.94 | 92.80 | 92.64 | 93.37 | 93.47 | 93.61 | 93.52 | 93.58 | 93.76 | 93.22 | 93.32 |
| | N | 4454 | 4025 | 4458 | 4315 | 3533 | 4317 | 4462 | 4463 | 4320 | 4464 | 4320 | 4464 |
| Soil water content (mm³) | MEAN | 0.17 | 0.19 | 0.20 | 0.22 | 0.22 | 0.20 | 0.17 | 0.17 | 0.17 | 0.16 | 0.15 | 0.16 |
| | MEDIAN | 0.17 | 0.18 | 0.20 | 0.22 | 0.22 | 0.20 | 0.16 | 0.17 | 0.17 | 0.16 | 0.15 | 0.16 |
| | SD | 0.02 | 0.01 | 0.01 | 0.01 | 0.01 | 0.02 | 0.02 | 0.02 | 0.02 | 0.02 | 0.01 | 0.02 |
| | MAX | 0.31 | 0.22 | 0.25 | 0.28 | 0.29 | 0.33 | 0.31 | 0.30 | 0.22 | 0.33 | 0.17 | 0.30 |
| | MIN | 0.15 | 0.17 | 0.19 | 0.20 | 0.20 | 0.17 | 0.14 | 0.15 | 0.15 | 0.14 | 0.14 | 0.14 |
| | N | 4459 | 4025 | 4458 | 4315 | 3533 | 4317 | 4462 | 4463 | 4320 | 4464 | 4320 | 4464 |
| Soil Temperature (°C) | MEAN | 22.89 | 23.12 | 23.37 | 23.15 | 23.80 | 25.27 | 26.07 | 26.18 | 25.58 | 24.56 | 23.48 | 23.16 |
| | MEDIAN | 22.80 | 23.10 | 23.40 | 23.20 | 23.60 | 25.30 | 26.10 | 26.20 | 25.50 | 24.60 | 23.40 | 23.10 |
| | SD | 0.27 | 0.19 | 0.07 | 0.13 | 0.49 | 0.21 | 0.24 | 0.13 | 0.25 | 0.30 | 0.40 | 0.18 |
| | MAX | 23.40 | 23.40 | 23.50 | 23.30 | 24.60 | 25.70 | 26.40 | 26.40 | 26.10 | 25.10 | 24.31 | 23.50 |
| | MIN | 22.20 | 22.60 | 23.00 | 22.70 | 23.20 | 24.90 | 25.50 | 25.90 | 25.10 | 24.10 | 22.90 | 22.30 |
| | N | 4459 | 4026 | 4459 | 4316 | 3533 | 4317 | 4462 | 4463 | 4320 | 4464 | 4320 | 4464 |
| Conductivity (mScm) | MEAN | 1.55 | 1.79 | 2.00 | 2.27 | 2.61 | 1.32 | 1.22 | 1.04 | 1.41 | 1.49 | 1.34 | 1.54 |
| | MEDIAN | 1.49 | 1.69 | 1.97 | 2.26 | 2.57 | 1.32 | 1.20 | 1.04 | 1.38 | 1.47 | 1.34 | 1.46 |
| | SD | 0.14 | 0.14 | 0.11 | 0.14 | 0.18 | 0.18 | 0.16 | 0.10 | 0.09 | 0.09 | 0.01 | 0.18 |
| | MAX | 2.61 | 2.18 | 3.06 | 3.35 | 3.67 | 2.25 | 2.33 | 2.21 | 2.15 | 2.18 | 1.37 | 2.17 |
| | MIN | 1.31 | 1.66 | 1.89 | 2.06 | 2.32 | 0.97 | 0.94 | 0.89 | 1.30 | 1.35 | 1.32 | 1.36 |
| | N | 4122 | 4025 | 4458 | 4315 | 3533 | 4317 | 3059 | 4025 | 4071 | 2966 | 2009 | 3140 |

N = number of samples collected by the sensor (one sample every 10 min).

Spearman's rank correlation coefficient ($r_s$) varied greatly when comparing the different environmental parameters measured by the ATMOS14 and TEROS12 probes. The correlation between the different environmental parameters measured by ATMOS14 and TEROS12 probes is presented in Table 3. Principal component analysis (PCA) is represented in Figure 7 and shows the correlation between air temperature and soil temperature, as well as between soil conductivity and soil water content. Principal component analysis also showed that the variables air temperature, soil temperature, soil conductivity, and soil water content were responsible for most of the variance in the data, while ambient pressure and relative humidity had a smaller contribution to this variance. As it can be appreciated in Table 4, principal component 1 only represented 38.82% of the total variance, principal component 2 only represented 24.26% of the variance, and principal component 3 only represented 17.325% of the variance. If we use the correlations between the principal components and the original variables to interpret these principal components (Table 5), we found that principal component 1 was strongly correlated with the variables soil temperature, air temperature, and soil conductivity; principal component 2 was strongly correlated with atmospheric pressure and soil water content; and principal component 3 was strongly correlated with relative humidity.

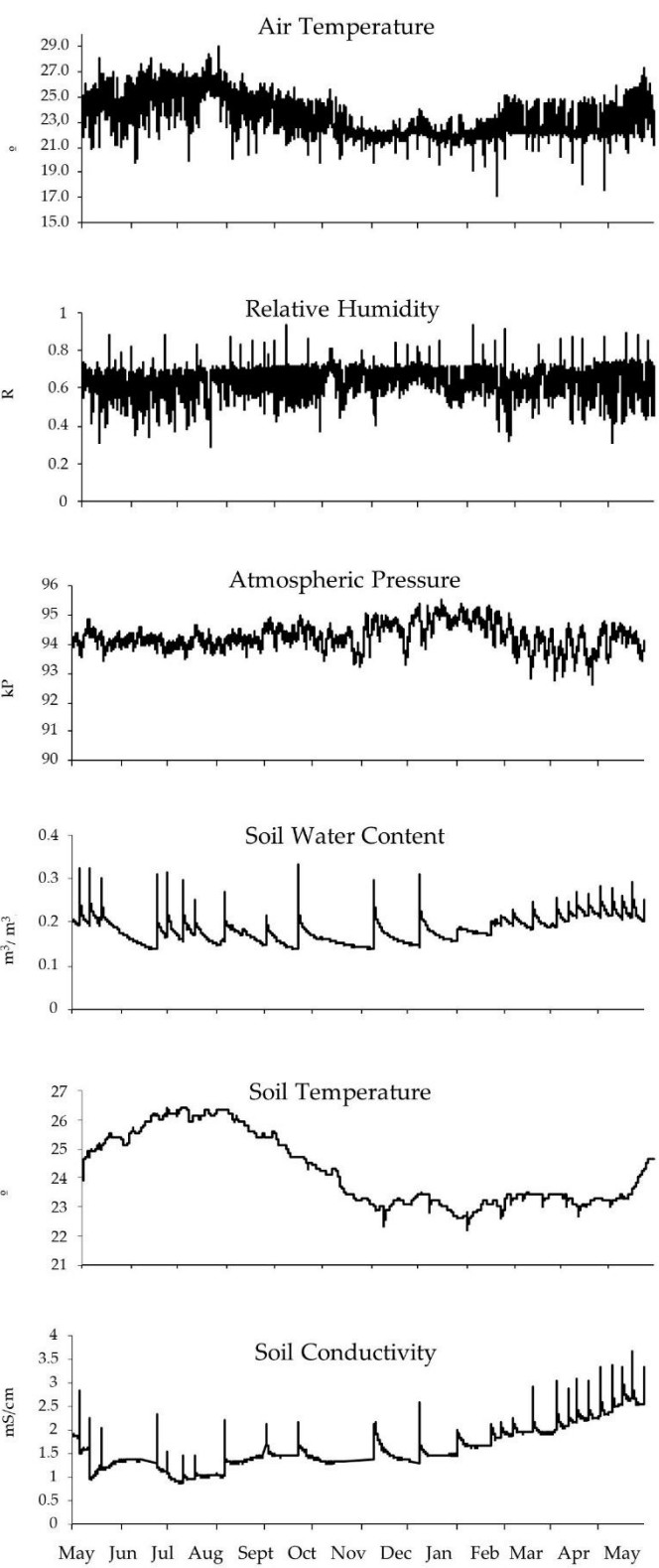

**Figure 6.** Values for the main environmental parameters recorded every 10 min by ATMOS14 and TEROS12 probes from 25 May 2021 to 27 May 2022.

**Table 3.** Spearman's rank correlation coefficient ($r_s$) between the six environmental parameters measured by the ATMOS14 and TEROS12 sample probes (conf. level = 0.95).

| Variable | RH | KPA | MMWC | SOILTEMP | MSCM | AIRTEMP |
|---|---|---|---|---|---|---|
| RH | 1.000 | −0.043 | −0.037 | −0.144 | 0.108 | −0.312 |
| KPA | −0.043 | 1.000 | −0.156 | −0.186 | 0.073 | −0.264 |
| MMWC | −0.037 | −0.165 | 1.000 | −0.117 | 0.572 | −0.042 |
| SOILTEMP | −0.144 | −0.186 | −0. 117 | 1.000 | −0.652 | 0.758 |
| MSCM | 0.108 | 0.073 | 0.572 | −0.652 | 1.000 | −0.389 |
| AIRTEMP | −0.312 | −0.264 | −0.042 | 0.758 | −0.389 | 1.000 |

RH, relative humidity; KPA, atmospheric pressure; MMWC, soil water content; SOILTEMP, Soil temperature; MSCM, soil conductivity; AIRTEMP, air temperature.

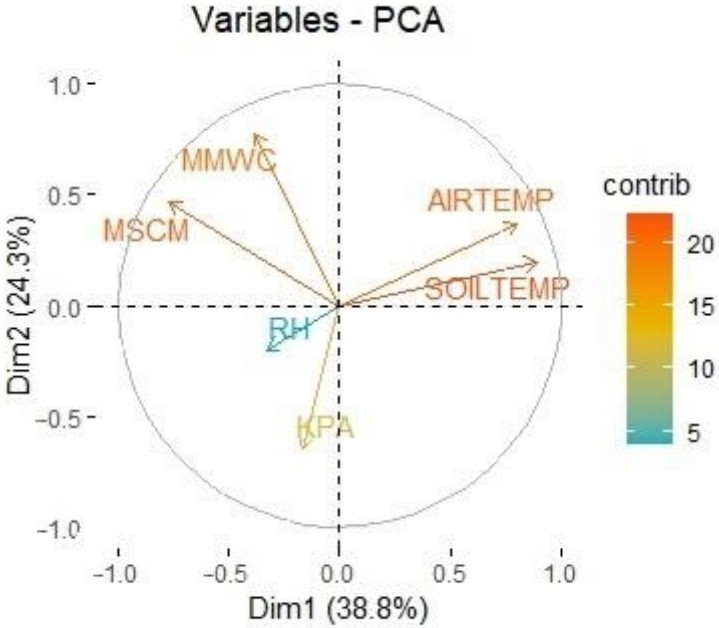

**Figure 7.** Principal component analysis (PCA) of the six different variables measured by the ATMOS14 and TEROS12 sample probes: AIRTEMP, air temperature (°C); SOILTEMP, soil temperature (°C); MSCM, conductivity (mS/cm); MMWC, soil water content (m³/m³); RH, relative humidity (RH); KPA, atmospheric pressure (kPa). *X–Y* axis corresponds to principal component 2 (Dim 2)—principal component 1 (Dim 1). The different colors represent the percentage of contribution (contrib) of each variable (environmental parameter) to data variance.

**Table 4.** Principal component analysis of the six environmental parameters measured by the AT-MOS14 and TEROS12 sample probes: variance analysis of each principal component.

| | Principal Component | | | | | |
|---|---|---|---|---|---|---|
| | 1 | 2 | 3 | 4 | 5 | 6 |
| Variance | 2.330 | 1.455 | 1.040 | 0.647 | 0.347 | 0.181 |
| % of variance | 38.826 | 24.257 | 17.325 | 10.790 | 5.779 | 3.022 |
| Cumulative % of variance | 38.826 | 63.083 | 80.408 | 91.199 | 96.978 | 100.000 |

As it can be appreciated in Figure 8, soil temperature was always superior to air temperature. Statistically significant differences ($p < 0.01$; Mann–Whitney U test) were detected between soil and air temperature for all studied months, except for May ($p = 0.027$).

**Table 5.** Analysis of the three principal components of the PCA for the six environmental parameters measured by the ATMOS14 and TEROS12 sample probes.

| Variable | Principal Component | | | | | |
| | 1 | | 2 | | 3 | |
| | Cor | Contr (%) | Cor | Contr (%) | Cor | Contrib(%) |
|---|---|---|---|---|---|---|
| RH | −0.331 | 4.698 | −0.204 | 2.871 | 0.824 | 65.344 |
| KPA | −0.168 | 1.218 | −0.647 | 28.727 | 0.537 | 27.771 |
| MMWC | −0.381 | 6.233 | 0.777 | 41.489 | 0.161 | 2.492 |
| SOILTEMP | 0.897 | 34.541 | 0.200 | 2.737 | 0.156 | 2.334 |
| MSCM | −0.770 | 25.469 | 0.464 | 14.781 | −0.093 | 0.838 |
| AIRTEMP | −0.805 | 27.840 | 0.370 | 9.394 | 0.137 | 1.222 |

RH, relative humidity; KPA, atmospheric pressure; MMWC, soil water content; SOILTEMP, Soil temperature; MSCM, soil conductivity; AIRTEMP, air temperature; Cor, correlation; Contr, contribution.

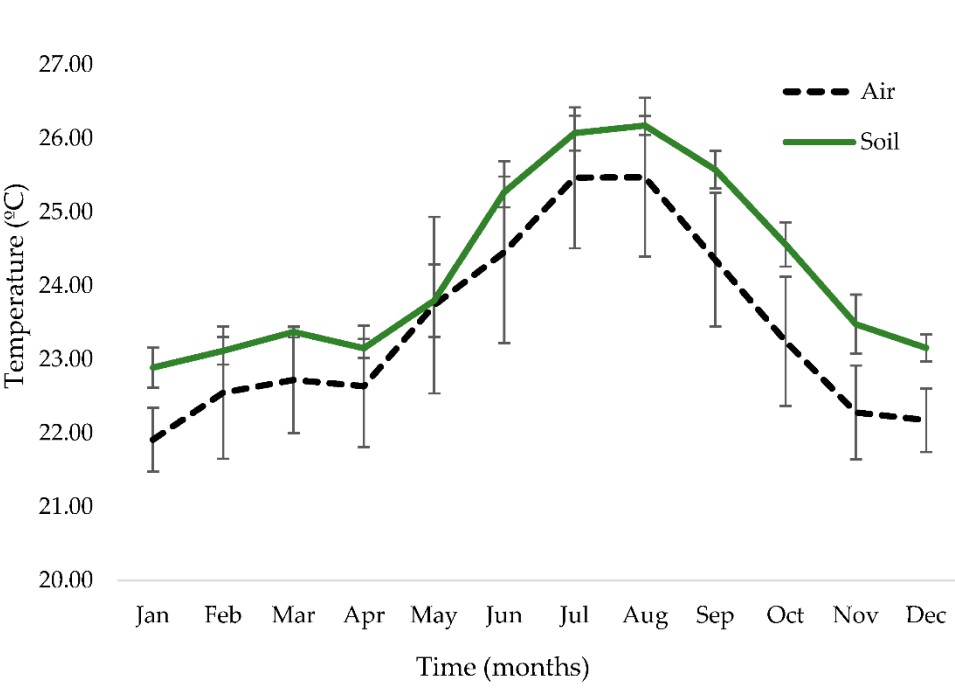

**Figure 8.** Mean ± SD air and soil temperatures at Biodomo—Parque de las Ciencias de Granada. Statistically significant differences ($p < 0.01$; Mann–Whitney U test) were detected between air and soil temperature in all months except May ($p = 0.027$).

## 4. Discussion

This study shows the potential of a detailed and accurate monitoring of the environmental parameters such as light intensity, air temperature, relative humidity, atmospheric pressure, soil temperature, soil water content, and soil conductivity. Our manuscript presents an affordable methodology that can provide the institution with important data. It also shows that many of the environmental parameters are correlated, demonstrating the potential use of the obtained information for improving ambient conditions and the efficient selection of vegetable species to form part of a zoological and botanical garden.

In our study, statistically significant differences were detected for annual DLI measurements not only in different animal enclosures within the building, but also between sampling spots in the same enclosure (For instance, between the sampling spots "Madagascar enclosure spot 1" vs. "Madagascar enclosure spot 2" and "Amazon rainforest spot 1" vs. "Amazon rainforest spot 2", as well as "Amazon rainforest spot 1" vs. "Amazon rainforest spot 3" ($p < 0.05$). These significant variations reveal the importance of developing studies determining DLI in the main areas where vegetation is located at the facility, to allow the efficient selection and location of the vegetable species within the enclosure depending on

their DLI requirements. The DLI quantification allows us to accurately determine the most suitable location within the enclosure for the selected species, facilitating its adaptation and development [9,12,14]. Furthermore, quantifying DLI allows the correction of light input if DLI values are not the ones desired, which can be increased by adding supplemental light or reduced by providing shade. This is important, as a linear relationship has been found between plant growth and the cumulative DLI, being an accurate indicator for growth and development [23].

The differences between sampling spots and the possibility to determine DLI levels at the desired study locations within the enclosure facilities allows the detection of DLI variations which could be produced by the building and enclosure design, which may not be appreciable to the naked eye [8,24]. The use of the method described in this study can allow the detection of the anomalies caused by the aesthetics of the building to be detected and corrected, perhaps by placing an artificial light source for the months in which the design of the building and the incidence of the sun produce shadows. An example of the detection of these anomalies can be observed at sampling point 1 (Asiatic island): during the months of October, November, and December, the sun's incidence makes the sun rays directly hit the island through some windows, producing a significant increase in DLI measurements during those months, with higher DLI levels than those recorded during the summer months. This is because when the sun is higher, the building structure creates a shadow in this location, reducing DLI levels. This information is of great use, as it will allow the technical and gardening team at Biodomo to place a PAR LED panel that will supplement this location of the building during the months when the structure creates this shadow, to achieve the desired DLI.

The implementation of studies such as the one proposed by the authors will allow a more efficient use of artificial light, as artificial light panels will only be used in areas where DLI are under the desired levels, promoting energy saving. This allows the maximum use of sunlight, and the placement and use of artificial light panels only at the spot/season/hour required. The DLI data would allow a separation in the artificial light panels control and the design of a lighting system that would work "like clockwork", varying daily and between seasons, depending on the design of the building and the vegetable species' requirements. This would avoid having all the light panels turned on continuously, many of them probably even located in spots where DLI is already over the desired levels. This whole process would allow energy savings, as well as a more efficient, economic, and sustainable light management in zoological and botanical gardens.

The experimental design and the methodology proposed in this study, with the use of high precision sensors for measuring environmental variables, opens the possibility of future studies with animal species maintained in indoor facilities, which also have certain DLI and UV necessities [25,26]. The knowledge of environmental DLI at the indoor facility will allow the optimal selection of the most appropriate species for the environmental light levels, as well as determining if an increase or decrease in light intensity is necessary for the enclosure depending on the species housed. A detailed study of these parameters could help to make a better choice of animal species or a better adaptation of the facility for the existing animals, thus improving their health and well-being [25,27]. Since the DLI sensor is submersible, its usefulness for designing correct lighting in aquatic environments for corals and aquatic plants in aquariums is significant, based on the fact that each species has certain DLI needs [12,28].

An important limitation of the light evaluation in this study was that only one DLI sensor was available for sampling, and therefore a rotating system was developed to determine the DLI levels in the selected spots within Biodomo. Future studies could include more than one sensor, which could be placed at the pre-determined sampling locations and therefore provide continuous DLI data for every sampling spot without the need to rotate the sensor, reducing sampling time and avoiding errors produced by the daily cloud and DLI variations.

When focusing on relative humidity, despite statistically significant variations being detected for this parameter across the year in our study, no statistically significant differences were detected for air relative humidity between August (which is one of the warmest and driest months in Granada, Spain) and February (which is one of the coldest months and accumulates most of the precipitation in Granada) [29]. The absence of statistically significant differences between these months can be explained as the environmental humidity control system (described in the Methodology section and based on automated air conditioning and fog systems at Biodomo) maintaining a relatively consistent relative humidity at Biodomo. Previous studies revealed that the maintenance of constant and appropriate relative humidity levels is essential not only for optimal plant development (as it affects photosynthesis, leaf growth, and disease incidence) but also for guaranteeing animal health and welfare [30,31]. Because of this, the monitorization of air relative humidity across the year in indoor facilities maintaining animal and plant species is of great importance to provide the most appropriate environmental conditions and avoid the occurrence of diseases.

Despite the variations observed in the temperatures registered in our study across the year, the indoor air and soil temperature results, with a maximum of 29.10 and 26.20 °C, respectively, and a minimum of 17.15 and 22.20 °C, respectively, are relatively stable compared to the continental climate of Granada, which in 2021 had maximum temperatures of 45 °C during August and minimum temperatures of −5 °C during February, revealing the efficiency of the automated climate control system design at Biodomo [29]. The maintenance of a constant temperature with mild variations in this parameter is important, as zoological and botanical gardens frequently host tropical species highly sensitive to temperature variations. The presence of sensors continuously registering environmental temperature is important to detect possible variations in temperature, which may exceed the desired maximum and minimum limits, and together with variations in air humidity can lead to disease in the animal and plant collections [32,33]. One factor to consider when keeping plant and animal species at indoor facilities is the possible alteration of circadian rhythms, since we are creating an artificial environment where it would be difficult to replicate the cycles of light and temperature that occur in nature [34,35]. This problem is exacerbated in mixed facilities where we house species from such diverse locations under the same roof. This marks an important guideline in the species selection criteria, considering that alterations in these cycles can cause certain problems, not only in the animals but also in the plants, such as alterations in the immune system or changes in their morphology [36,37]. In the care and management of certain animal species where seasonal cycles are very important, being able to emulate these cycles would be essential for their well-being and behavioral development [38].

The maintenance of plants in interior facilities such as indoor zoological gardens usually has particularities, with the limitation of space for the development of plants and their roots as an important concern. In Biodomo, all the plants are cultivated in buried pots, to avoid uncontrolled root proliferation, to have more precise control on irrigation, and to facilitate plant translocation. This study shows that the soil sensor can provide relevant information for daily maintenance and decision making, such as when to apply irrigation correctly. This agrees with previous studies evaluating the potential use of soil sensors to define efficient irrigation schedules [39].

This study shows that soil temperature and air temperature, while being strongly correlated ($r_s = 0.758$), differ significantly from each other throughout the year, being soil temperature significantly higher at the sampled spot when compared to air temperature in all months except May. Biodomo does not count with a soil heating and refrigeration system, and the soil, due to its solid condition, accumulates thermal energy [40]. As can be observed in Figure 8, during spring the air temperature increases rapidly, reaching soil temperature in May, and increasing both soil and air temperatures sharply during the summer months, being that soil temperature is always higher due to solar radiation and energy accumulation. At the end of August, both air and soil temperatures drop, maintaining soil always at a

higher temperature during autumn and winter. The determination of soil temperature is of great interest, as previous studies already demonstrated that these parameters directly affect the growth of the root system, and therefore the optimal development of the plant [41].

When looking at conductivity, our study detected a marked relationship between conductivity and soil water content (with a correlation of $r_s = 0.572$). Furthermore, Figure 6 allows the visual detection of different peaks in the graphs for both parameters, which is related to the irrigations carried out throughout the year, observing a higher irrigation frequency during the months of March to August and a lower frequency from August to February. The live determination of soil parameters such as temperature, water content, and conductivity will also allow the early diagnosis and prevention of fungal pathologies since there is a direct relationship between these parameters and the appearance of fungal diseases [42]. A limitation to this study is that only one TEROS12 soil sensor was available during sampling, and therefore we could only determine soil parameters for a fixed location at Biodomo. Future studies with soil sensors placed across different locations will allow a more precise evaluation of soil conditions and if plant requirements are met at the different enclosures. This will allow the design of a detailed and sustainable irrigation schedule. For instance, this efficient irrigation system based on independent soil sensors would provide water only in the locations with a water deficit, always guaranteeing the optimal soil conditions and promoting plant development while saving water.

By relating atmospheric pressure and relative humidity we can obtain another useful parameter for plant management: evapotranspiration. This parameter allows the calculation of the amount of water that is being lost both by the plants and the soil [43]. Evapotranspiration, together with the data obtained by a soil probe, allows us to accurately and reliably determine plant water needs and establish more efficient irrigation schedules [43,44]. The objective of this study was not to present in detail the differences in atmospheric pressure and therefore in evapotranspiration, but to show an effective methodology to determine these parameters, which vary significantly over time. Future studies determining the variations in atmospheric pressure in indoor and outdoor zoological facilities will allow a more adequate interpretation of these parameters and their usefulness beyond greenhouse crops.

Together with the previously described correlations between variables, the PCA analysis for the six environmental parameters recorded by the fixed sensors (ATMOS14 and TEROS12) showed that air relative humidity contributed very poorly to the data variance, atmospheric pressure contributed moderately, while soil water content, soil conductivity, soil temperature and air temperature contributed greatly to data variance (Figure 7). As can be appreciated in Table 4, even when evaluating the principal components with a variance over 1.000 (principal components 1, 2, and 3), they only represent 80.408% of the total variance, being a limitation of the PCA analysis in our study. It should also be considered that data variance, and the contribution of each variable to the principal components, could vary greatly depending on the conditions of each facility. In our study, the results of the PCA analysis could have been influenced by the climate control tools available at Biodomo: air relative humidity was strictly regulated by a climate control console linked to a fogging system, reducing data variance. Atmospheric pressure was not regulated, and its variance could significantly differ between studies. Despite air temperature being regulated automatically by a climate control system, this variable, together with the rest of environmental parameters studied, were greatly dependent on the daily management of the facility and the non-automated environmental control elements previously described in the methodology section of this manuscript. Although this was not the objective of our study, further trials could be performed evaluating in detail how the different auxiliary environmental parameters control tools affect each of the measured variables, so that the use of these elements can be optimized.

Together with the evaluation of environmental parameters described in this manuscript, further environmental studies performed in zoological and botanical gardens could benefit from the determination and quantification of both natural and anthropogenic air pollu-

tants, particularly particulate matter [45]. These complementary studies will assist in the evaluation of indoor air quality, as indoor pollution has been shown to have severe health impacts and should be considered, especially in those facilities built near large urban areas. In addition, modern indoor zoological and botanical gardens such as Biodomo should operate with air treatment units equipped with air filters, which help control these harmful particles.

Ideally, desired environmental parameters should be strongly considered before constructing the building that will host the animal and plant collections, as it will facilitate the posterior maintenance of the optimal ambient conditions [24]. Once the facility is built, the development of environmental studies, such as the one proposed in this article, will allow determination of the effectiveness of the design and the need to implement artificial light sources, or modify the rest of the environmental conditions. Further studies determining environmental parameters such as DLI in facilities housing delicate animal species which are highly dependent on their environment such as corals, fish, reptiles, amphibians, and many avian and mammal species will provide important information for their successful care and management.

## 5. Conclusions

This study provides a simple and efficient method for the evaluation of environmental parameters in zoological and botanical gardens housing plants and animal species in indoor facilities. Studies determining environmental parameters are frequently developed in livestock and horticultural production facilities, while this type of studies is still uncommon in zoological/botanical facilities. This study applies a combination of the environmental monitorization systems used in plant and animal production facilities, for the design and improvement of climate management in indoor zoological and botanical gardens. The detailed methodology description provided in this study can be useful in modern zoological facilities, in which the optimal development and welfare of the species maintained is increasingly important. Results provided in this study show the importance of each parameter determination for the optimization of the facility design, species selection and location.

**Author Contributions:** Conceptualization, P.M.-E and L.L.C.-P.; methodology, L.L.C.-P. and P.M.-E.; software, L.L.C.-P. and P.M.-E.; formal analysis, L.L.C.-P. and P.M.-E.; investigation, L.L.C.-P. and P.M.-E.; data curation, P.M.-E.; writing—original draft preparation. L.L.T.-P. and P.M.-E.; writing—review and editing. L.L.C.-P., P.M.-E and J.G.-M.; supervision, P.M.-E. All authors have read and agreed to the published version of the manuscript.

**Funding:** Sensors used in this study were funded by Rain Forest S.L.

**Institutional Review Board Statement:** Not applicable.

**Data Availability Statement:** Data provided in this study will be made available upon request to the authors.

**Acknowledgments:** The authors would like to acknowledge the help provided by the team of professionals working at Biodomo—Parque de las Ciencias de Granada.

**Conflicts of Interest:** The authors declare no conflict of interest. The funders had no role in the design of the study; in the collection, analyses, or interpretation of data; in the writing of the manuscript or in the decision to publish the results.

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
