# Peer review of "A Model for Accurate Determination of Environmental Parameters in Indoor Zoological and Botanical Gardens Supporting Efficient Species Management"

_2673-5636, doi:10.3390/jzbg3040038_

Round 1

Reviewer 1 Report

The reviewer has no complaints about the content, data processing and quality of the studies. However, there is an unaccounted factor that is fundamentally important for both plants and animals - the natural circadian rhythms of thermal radiation and thermal balance, which cannot be created in protected greenhouse conditions. In nature, plants use the night period to reduce their temperature below the dew point, i.e. condense water on the surface. Not being a zoologist, I will not write about the rhythms of animal behavior, which are affected by the nightly cooling of the soil and air, but I am sure that such a connection exists. Thus, keeping animals and plants under the dome of greenhouses is convenient only for visitors to such facilities, but unusual for their inhabitants. No matter how temperature, humidity and light are regulated - all these devices will not replace the starry sky overhead, through which the Earth's surface loses its heat.

Author Response

Dear Reviewer,

Thank you very much for the important commentary. Of course, the circadian rhythms produced by both light and temperature could definitely be disrupted in a greenhouse. However, the maintenance of tropical species in places where the temperature drops require greenhouses or heated facilities. A paragraph (Lines 541-551) has been added specifying this limitation of greenhouses and stating that it should be considered in environmental studies performed in indoor zoological and botanical gardens.

Reviewer 2 Report

Some minor corrections/comments/suggestions can be found in the attached file.

Reviewer 3 Report

Dear Authors and Editors,

The manuscript " A Model for Accurate Determination of Environmental Parameters in Zoological and Botanical Gardens Supporting Efficient Species Management" by León Latif Corral-Pesquerai et al. Proposed evaluation of environmental parameters for the optimal selection and location of vegetable species not only in vegetable production facilities and greenhouses, but also in zoological and botanical gardens. This article is of some importance, but it does not provide much insight into how the models were selected, and some parts are difficult to follow. I would be willing to re-evaluate this manuscript with a major revision.

 1.  The expression logic in the abstract is quite confusing, and does not show the methodology and highlights the significance finding of the article.

2.  The flow of the aforementioned parts in the introduction is challenging to follow. I will suggest that it should begin with the current climate situation relating to plants/vegetation from a scientific and linguistic perspective.

3. I did not see the study's aims and objectives, nor did I see the research questions. I will suggest including that at the end of the introduction.

4. Various models can be used for such studies, including Maxent and PCA.

5. The analysis of correlation can be very complicated to understand and needs to be redrawn.

6.   Fig 5. Explanation should be more detailed.

7. Please provide the table of the significant difference among the different environmental variables .

8,  The conclusion is too simple. In conclusions, “Our newly proposed cost-effective, and efficient method for the evaluation of environmental parameters in zoological and botanical gardens housing plants and animal species in indoor facilities.” Why do the methods in the manuscript have these characteristics and what methods are they compared? This should be summarized in depth.

9. Must cites recent paper from JZBG.

Round 2

Reviewer 3 Report

Dear Authors!

Overall, this version is much better and cleaned up. While I agree that is is acceptable for publication. I think with some minor revision, including adding "Implication and evaluations of indoor soot particles from domestic fuel energy sources using characterization techniques in northern Pakistan".

I recommended for Publication.

Author Response

Dear Reviewer, 

Thank you for your suggestion. We have included a brief statement (Lines 622-630) in the manuscript referring to your last remarks. We have also included the suggested reference.